# Early-life immunological and microbial differences between East African and North European children
Noora Nurminen [1], Yue-Mei Fan [1], Emma Kortekangas[1], Jake Lin [1,2], Lotta Hallamaa[1], Kenneth Maleta[3], Kirsi-Maarit Lehto [1,4], Olli H. Laitinen[1], Aki Sinkkonen[5], Johanna Lempainen [6,7,8], Jorma Toppari [7,9,10,11], Riitta Veijola [12], Kalle Kurppa[1,13,14], Mikael Knip [13,15], Ulla Ashorn[1], Sami Oikarinen[1], Per Ashorn [1] & Heikki Hyöty [1,16] ✉

## Abstract

**Background** The urbanization of African populations adopting Westernized lifestyles might be connected to changes in microbial exposure and immune system activity that are harmful to health and increase the risk of non-communicable diseases, such as immune-mediated diseases. The study aims to compare microbial exposure, immune system markers, and gut microbiota between rural African and Westernized Northern European children to delineate whether there are differences present in these factors in early childhood.
**Methods** We compared innate immune cytokines in plasma (IL-10, IL-6, IL-1β, and TNF-α) using Luminex, gut microbiota using 16S rRNA sequencing, and microbial infections using qPCR in early childhood longitudinal sample series of children from rural Africa (participants from the iLINS-DYAD-M study conducted in Malawi) and from Northern Europe (participants from the DIPP study conducted in Finland) to identify differences which could be associated with negative health outcomes in Westernized societies.
**Results** Here, we show that the levels of plasma cytokines and frequency of stool pathogen positivity are substantially higher in Malawian than in Finnish children and that some of the cytokines differ in their longitudinal pattern between the two groups. Also, the diversity and composition of gut microbiota differ between the groups at the age of 6 months and diverge more with increasing age.
**Conclusions** These results highlight the early emergence of differences in the immune system and gut microbiota between children living in extremities of the microbial exposure gradient. These differences add to the existing knowledge of possible factors contributing to increasing prevalence of chronic inflammatory diseases in African societies shifting towards more Westernized lifestyle.

## Plain language summary

A person's immune system defends the body from infections, such as those caused by microbes. Many diseases that are linked to the functioning of the immune system, such as allergies, are more common in Westernized populations than in those with more traditional lifestyles, such as those found in rural Africa. This increased disease risk might be connected to reduced microbial exposure, particularly in early childhood. In this study, we examined markers of immune system function from blood samples, as well as gut bacteria, viruses and parasites in stool samples from young children living in rural Africa and Northern Europe. Our findings reveal that gut microbes and the immune system of these children are very different at 6 months of age. This might be linked to the development of immune-mediated diseases later in life. This study highlights the importance of early life exposures to the development of the immune system, and could provide useful information about how best to prevent the development of immune system diseases later in life.

The prevalence of immune-mediated diseases such as allergic diseases, type 1 diabetes, and inflammatory bowel diseases is higher in high-income countries and populations with urban lifestyles than in low-income countries and populations with more traditional lifestyles[1–3]. The prevalence of these diseases seems to be increasing also in African societies, shifting towards a more Westernized urban lifestyle characterized by reduced contact with environmental microbial exposures, less close contact with animals, a more sedentary lifestyle, and a shift to a diet richer in processed

foods[4–6]. The biodiversity hypothesis suggests that a normally functioning immune system requires an early high-diversity microbial exposure for its development and maturation towards self-tolerance and efficacious protection against pathogens—lack of these exposures leads to changes in the immune system and susceptibility to immune-mediated diseases[7].

The first 1000 days of life have been postulated to be a critical window for the maturation of the immune system and the establishment of the gut microbiota[8,9]. At the same time, the pathogenetic process of several

immune-mediated diseases begins already during this period[10,11] and differences in microbiota have been associated with several of these diseases[12–15]. Recent studies have revealed that the main events of immune system imprinting might be mediated through the microbiome, highlighting the early influence of the infant's microbial exposure[16]. It could thus be hypothesized that early-life microbial exposure could modulate the maturating microbiota and immune system and impact the development of immune-mediated diseases. Despite this possible connection, little research is done where early-life immune status, microbiota and microbial infections are analyzed simultaneously and consecutively in longitudinally followed children who live in contrasting microbial environments.

Using longitudinal analysis and machine learning, the current study aims to explore overall and temporal differences in innate immune cytokines, pathogenic burden, and gut microbiota in early childhood between European and rural African populations representing environments with low and high microbial exposure. The rural African children have been born and are residing in rural Malawi whereas the European children have been born and are residing in Finland. The prevalence of several immune-mediated diseases is notably higher in Finland than in Malawi: age-standardized prevalence of rheumatoid arthritis is approximately five times higher, whereas the prevalence of atopic dermatitis is about four times higher[17]. We find clear differences between these groups in immune markers, gut microbiota and pathogen positivity and associations between stool microbes and blood cytokines are also found. Our findings emphasize the earliness of differences in the immune system and gut microbiota in environments contrasting in microbial exposure.

## Methods

### Study subjects and samples

This study includes blood and stool samples and anthropometric measurements of 40 participants of iLiNS-DYAD-M dietary intervention trial (clinicaltrials.gov identifier NCT01239693) conducted in rural Malawi (Table 1). The primary and secondary outcomes of the study have been published elsewhere[18,19]. The enrollment took place in one public district hospital (Mangochi), one rural semiprivate hospital (Malindi), and one rural public health center (Lungwena) in Mangochi District, Southern Malawi. The families were recruited 2011–2015 and the data used in this study was collected 2012–2015. The children were followed-up with anthropometric assessment and stool and blood sample collection at 6, 18, and 30 months of age. Samples were taken according to a predetermined schedule instead of the perceived clinical symptoms of infections. Venous blood samples were collected into heparin-treated tubes and centrifuged at 2000 x $g$ for 15 min by a trained lab technician to obtain plasma, which was aliquoted into cryovials and stored at −80 °C. One aliquot was later shipped on dry ice to Tampere University in Finland for analysis. Stool samples were collected at home by the mothers who were trained to collect stool samples from participating children and provided with sample collection containers a day before the scheduled visit. After collection, stool samples were sealed, labeled, and stored in Ziploc bags on frozen ice packs in cooler bags. The samples were transported to local laboratories, where they were aliquoted by a lab technician and temporarily stored at −20 °C before being transferred to the central laboratory in Mangochi for storage at −80 °C — all within 48 h. If the child had diarrhea at home visit, the stool collection was postponed for two weeks. Samples were collected between May 2012 and March 2015. Half of the Malawian children were stunted, but since the immunological markers and gut microbial composition did not differ between Malawian stunted and non-stunted children in this analysis, we combined them for further analyses (Table 1). Ethical approval for iLiNS-DYAD-M was obtained from the College of Medicine Research Ethics Committee, University of Malawi, and the Ethics Committee of Pirkanmaa Hospital District. Only participants whose caregivers gave informed consent were enrolled in the study.

The 40 Finnish children included in the current analysis were participants of the Finnish Diabetes Prediction and Prevention (DIPP) study

**Table 1 | Demographic characteristics and sample information of the participating children**

| | Finland (*N* = 40) | Malawi (*N* = 40) | *P*-value |
|---|---|---|---|
| Sex (female) | 16 (40.0%) | 15 (37.5%) | 1.000 |
| Mean WAZ (SD) | 0.64 (0.86) | −1.29 (1.50) | 2.622e-7* |
| Missing | 8 (5.0%) | 1 (0.8%) | |
| Stunted[a] | 0 (0.0%) | 20 (50.0%) | 7.798e-8 |
| Missing | 0 (0.0%) | 0 (0.0%) | |
| Breastfed at 18 months | 4 (11.1%) | 38 (97.4%) | 3.065e-13* |
| Missing | 4 (10.0%) | 1 (2.5%) | |
| **Source of drinking water** | | | |
| Piped water | - | 5 (12.5%) | |
| Borehole | - | 27 (67.5%) | |
| Well | - | 2 (5.0%) | |
| Lake | - | 5 (12.5%) | |
| Other | - | 1 (2.5%) | |
| **Sanitary facility** | | | |
| None | - | 4 (10.3%) | |
| Pit latrine | - | 35 (89.7%) | |
| Missing | - | 1 (2.5%) | |
| **Plasma samples** | | | |
| 6 months | 40 (100.0%) | 34 (85.0%) | |
| 18 months | 40 (100.0%) | 38 (95.0%) | |
| 24 months | 25 (62.5%) | - | |
| 30 months | - | 40 (100.0%) | |
| 36 months | 20 (50.0%) | - | |
| **Stool samples** | | | |
| 6 months | 40 (100.0%) | 40 (100.0%) | |
| 18 months | 40 (100.0%) | 40 (100.0%) | |
| 30 months | 25 (62.5%) | 40 (100.0%) | |

Mean WAZ is calculated for Finnish children, including 6-, 18-, 24- and 36-months data, whereas 6-, 18-, and 30-months data were used for Malawian children. Two-sided Mann–Whitney U test for continuous variables and Fisher's exact test for proportions.
WAZ Weight-for-age *Z*-score.
[a]Criteria for stunting: LAZ values below −2.
*missing values excluded from the analysis.

(clinicaltrials.gov identifier NCT03269084). The DIPP birth cohort study started in 1994 with screening for newborns for HLA susceptibility to type 1 diabetes and inviting families with a child carrying a risk HLA genotype to participate in prospective follow-up up to the age of 15 years. About 10% of the screened newborns fulfilled eligibility criteria for the long-term follow-up study[20]. The data included in this study were collected 1999–2013. The children included in the current analysis were confirmed to be islet-antibody negative to exclude any ongoing type 1 diabetes-related autoimmunity process. The children participated in regular follow-up visits with clinical examination and sample collection at 6, 18, 24, and 36 months of age at the Tampere University Hospital, Finland. Samples were taken according to a predetermined schedule and were not targeted according to clinical symptoms of infections. Blood samples were taken into Vacutainer® CPT™ Mononuclear Cell Preparation tubes with sodium citrate (BD Biosciences, USA), centrifuged according to the manufacturer's instructions for the separation of plasma. The plasma samples were stored at −80 °C. Stool samples were collected at home, mailed overnight and stored at −80 °C until analysis. Samples used in the current analysis were collected between May 1999 and August 2012. Since there were no identical blood sampling time point for both groups around 30 months of age, we included plasma samples taken at 24 months and 36 months of age from the Finnish study and 30-

month samples from the Malawian study. The guardians of all participating DIPP children gave written informed consent for genetic screening and follow-up. The study was approved by the Ethics Committee of Pirkanmaa Hospital District, Northern Ostrobothnia Hospital District, and the Hospital District of Southwest Finland. Both studies adhered to the principles of the Declaration of Helsinki.

## Anthropometric assessment and other data collection

For Malawian children, a trained anthropometrist measured the children's weight and length/height in triplicate, as described previously[18,19]. For Finnish children, a trained study nurse measured children's weight and length/height. The data on participants' sex and breastfeeding, and families' source of drinking water and sanitary facilities in Malawi were collected with structured data collection forms. Diet was assessed using a 7-day food frequency questionnaire form at 18 months of age in Malawi and a 3-day food frequency data collected at 12- and 24-months-of-age and published earlier from the DIPP study for Finland[21]. We calculated the percentage of daily consumers for each food category to harmonize the data. An average percent of daily consumers was calculated at 12- and 24-months-of-age for the Finnish children since there was no data at 18-months-of-age. We calculated age- and sex-standardized anthropometric indices—length-for-age $z$ score (LAZ), weight-for-age $z$ score (WAZ), and weight-for-length $z$ score (WLZ)—using the WHO Child Growth Standards[22] and considered LAZ values below −2 indicative of stunting.

## Multiplex cytokine assay

To analyze the activity of the immune system, the concentration of IL-1β, IL-6, IL-10, and TNF-α was analyzed from plasma samples using MILLI-PLEX® MAP kit (EMD Millipore) according to the manufacturer's instructions. Briefly, plasma samples were first incubated with antibody-coupled microspheres, then with biotinylated detection antibody and finally streptavidin–phycoerythrin was added. The captured bead complexes were measured with the Bio-Plex® 200 system (Bio-Rad Laboratories, Hercules, CA, USA) and data gathered using Bio-Plex Manager software (Bio-Rad Laboratories, Hercules, CA, USA, v. 4.1).

## Stool bacteriome analysis

Stool DNA was extracted using the PowerSoil DNA isolation kit (Qiagen, Hilden, Germany) with negative controls (PCR [polymerase chain reaction]-grade water) added into each extraction batch. Microbiome profiles were determined by next-generation sequencing of the V4 variable region of 16S ribosomal gene[23] and processed in a bioinformatic pipeline as described previously using the Mothur, Qiime, and Phyloseq tools[24]. The samples were rarified to 1326 sequences for comparison of bacterial diversity from an even sampling depth. Two samples were excluded due to too low a stool amount, and three samples were excluded due to low read count.

## Virus and parasite detection with PCR

RT-PCR was used for screening stool samples for adenovirus, enterovirus, norovirus, parechovirus, rotavirus, and rhinovirus. Stool samples were processed into 10% (w/v) suspension, nucleic acid was extracted using the modified Qiagen RNeasy96 kit (QIAGEN GmbH, Germany) and analyzed with real-time PCR using QuantiTect Probe kit (QIAGEN, Germany) according to the instructions on the Quantitect Probe kit using Taqman chemistry. Nucleic acid isolation and detection of parasites *G. lamblia* and *Cryptosporidium* species was performed using a method described earlier[25]. Briefly, a 10% (w/v) stool suspension was made using 0.2% bovine serum albumin in Hank's buffer. The samples were treated with a heat shock (10 min at 98 °C) and an overnight proteinase K treatment to disrupt the oocysts for nucleic acid isolation using QIAamp Viral RNA Mini Kit (Qiagen, Hilden, Germany). The sequences of primers and probes and the concentration of the oligonucleotides used in the qPCR reactions are listed in Supplemental Table 1. These viruses and parasites were selected because they are widely spread and reflect the hygienic conditions in which the children are living.

## Identification of the most important gut microbial factors for cytokines

To identify the most important gut microbial factors associated with each plasma cytokine, we first performed feature selection, based on importance and weight rankings, of genus-level bacterial abundances using the random forest method with default settings (randomForest R package). Subsequently, we included all genera that increased the mean squared error by at least 1%. Then the OTUs from the selected genera, along with bacterial diversity measures and pathogen PCR data, were combined and analyzed using the machine learning Elastic Net regularized regression method (glmnet R package[26]) to select the most important features associated with each of the plasma cytokines. Elastic net penalty (alpha) was set to 0.7, while the respective optimal models were selected using lambda.1se setting, which gives the most regularized model where the cross-validation error is within one standard error of the minimum validated errors. Finally, the selected variables from Elastic Net were added to separate multivariate regression models adjusted for sex, weight-for-age Z-score[22], batch, and breastfeeding.

## Statistics and reproducibility

All statistical analyses were carried out with the R software[27]. Two-sided Mann–Whitney U test was used for continuous variables and two-sided Fisher's exact test for counts and proportions. Bacterial community difference was analyzed using PERMANOVA and stool bacteriome was visualized using Krona[28]. Linear mixed-effects regression models (LMM, lme4 package) were applied to study longitudinal differences in cytokines and microbial alpha diversity between the groups. Country, age, breastfeeding, weight-for-age Z-score and interaction between age and country were included as fixed effects. Study subject was included as random intercept to account for repeated measures from the same participant. Cytokine concentrations were log2-transformed for feature selection, multiple regression and LMM models. Variables were standardized ($(x − μ)/σ$, where $μ$ is the mean value of the variable, and $σ$ is the standard deviation of the variable) for feature selection and multiple regression to facilitate comparison of the variables. Multiple regression results are presented as beta coefficients with 95% confidence interval (CI) and $P$ values. A $P$ value < 0.05 indicated statistical significance. The beta coefficients were interpreted as a standard change in the (log2) cytokine level as a result of a 1 SD change in the microbial variable. Sample sizes for blood and stool analyses are shown in Table 1. PCR analyses were conducted using three technical replicates, other analyses were done without technical replicates.

# Results

## Study population

40 children were included from both the Finnish and the Malawian study cohorts. Characteristics of study participants and number of samples at each age are summarized in Table 1. Malawian children had a lower weight-for-age Z-score, and they were more often stunted and breastfed at 18 months than Finnish children (Table 1).

## Plasma cytokine levels differ between Malawi and Finland in early childhood

Implying a higher activation of immune system during early childhood, Malawian children had higher mean plasma concentration of all measured cytokines (IL-10, IL-6, IL-1β, TNF-α) compared to Finnish children at 6 months of age (Fig. 1). At 18 months, only mean TNF-α concentration was higher among Malawian children whereas at 24–36 months of age mean IL-10, IL-1β, and TNF-α concentrations were higher in Malawian than in Finnish children (Fig. 1). Of note, due to differing sampling intervals in the oldest age group (30 months in Malawi vs. 24 and 36 months in Finland), the mean cytokine values in Malawian children at 30 months were also compared separately to those at either 24 months or 36 months in Finnish children. Malawian children had higher cytokine values in both comparisons (Supplemental Fig. S1). In addition, mean cytokine concentrations did not differ between 24- and 36-months of age among Finnish children.

**Fig. 1 | Plasma cytokine (IL-10, IL-6, IL-1β, and TNF-α) levels in Finnish and Malawian children.** The medians with interquartile ranges are indicated and differences are analyzed using two-sided Mann–Whitney U test. For Finnish children $n = 40$ and for Malawian children $n = 40$ biologically independent samples.

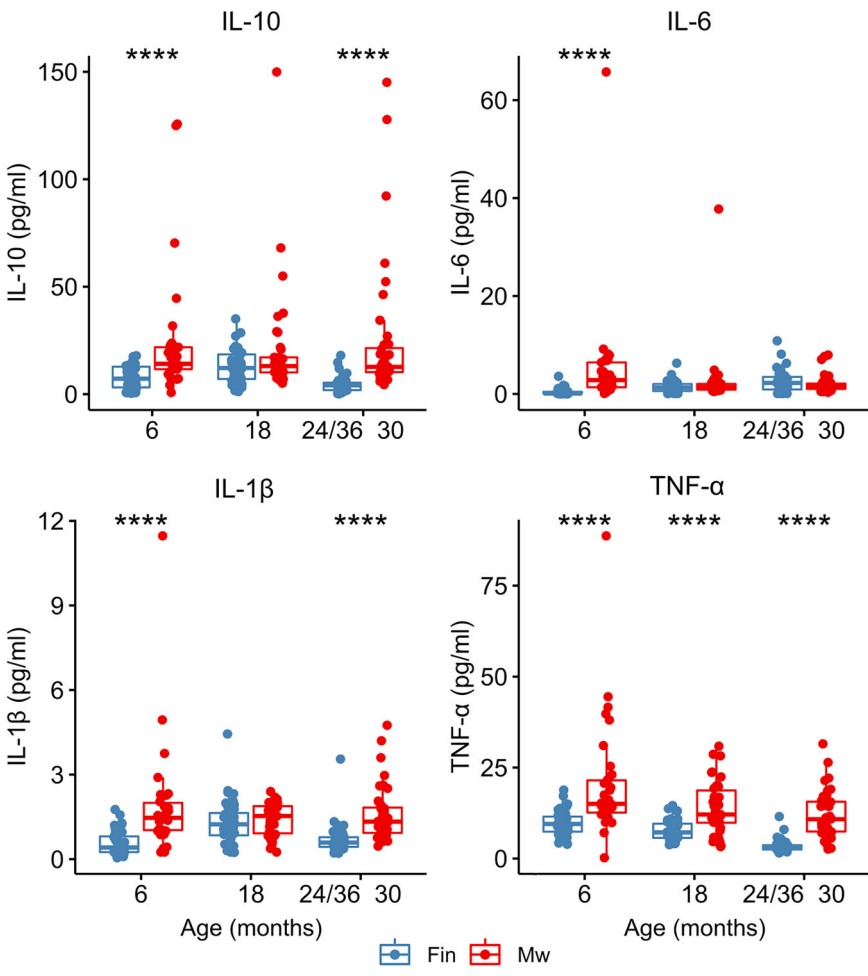

Assessment of the cytokine levels over time by LMM revealed differences between the Finnish and Malawian children in the trajectories of IL-10, IL-6 and TNF-α (Country:Age interaction term: $\beta = 0.067$, $P = 0.018$; $\beta = -0.174$, $P = 2.280E\text{-}07$, $\beta = 0.037$, $P = 0.039$, respectively, Supplemental Table 2 and Supplemental Fig. S2). When analyzing all samples together, plasma IL-6 concentration was lower in the males compared to the females (mean 1.7 vs. 3.4 pg/ml, $P = 0.032$), while plasma IL-10 (18.5 vs. 10.8 pg/ml, $P = 0.014$) and TNF-α concentrations (15.5 vs. 7.55 pg/ml, $P < 0.0001$) were higher if the child was breastfed during blood sampling.

## Gut bacterial diversity and community composition differ between Malawian and Finnish children

Gut bacterial alpha diversity differed between the countries. Chao1 index was higher at 6 and 30 months of age in the Malawian children, whereas Simpson diversity (1-D) was higher in Finnish children at 18 and 30 months of age (Fig. 2). Chao1 index emphasizes low-abundance taxa, whereas Simpson diversity gives more weight to dominant species. Other analyzed alpha diversity measures were higher in Malawian children (Supplementary Fig. S4). There was no difference between the Finnish and Malawian children in how alpha diversity (Simpson) changed over time (Supplementary Table 3 and Supplementary Fig. S4). The community composition measured by Bray–Curtis dissimilarity differed between the countries at every sampling age (PERMANOVA $P = 0.001$ for all sampling ages). The bacterial communities also differed within the countries between 6 and 18 months (PERMANOVA $P = 0.001$ both for Finland and Malawi) and 6 and 30 months (PERMANOVA $P = 0.001$ both for Finland and Malawi) (Fig. 2). Adding batch number as a covariate to the PERMANOVA analyses did not change the results. Stunting was not associated with community composition within Malawian children at any age (PERMANOVA $P = 0.922$,

$P = 0.602$, $P = 0.651$ for 6, 18, and 30 months of age, respectively). Visualizations of taxa distribution at each age in both groups are provided in Supplementary Fig. S3.

## Pathogens detected in stool samples of the Malawian and Finnish children

At every age (6, 18, and 30 months), adenovirus and enterovirus positivity were higher in Malawian stool samples compared to Finland (Fig. 3). Significantly higher positivity for parechovirus was observed in the Malawian children at 6 and 18 months of age and rhinovirus positivity at 6 and 30 months of age than in the Finnish children whereas norovirus was more common in the Malawian children at 6 months of age. *Giardia lamblia* and *Cryptosporidium* spp. were detected only in Malawian stools, where *G. lamblia* positivity was significantly higher at 18 and 30 months and *Cryptosporidium* spp. positivity at 6 months of age compared to Finnish children. No difference was found in rotavirus positivity between the Malawian and Finnish children. In addition, the Malawian children had a substantially higher frequency of multiple pathogen positivity in the same sample than the Finnish children (Supplementary Fig. S5).

## Microbes associated with plasma cytokine levels in Malawian and Finnish children at 6 and 18 months of age

To study the association between microbiome and pathogen positivity with systemic cytokine levels, we selected the most important microbial factors for each measured cytokine using a random forest algorithm and elastic net regularization and created adjusted linear regression models for these microbial factors. Separate models were created for both groups at 6 months (Fig. 4) and 18 months of age (Fig. 5) to account for the large differences observed in the gut microbiota between the groups and the dynamic changes

**Fig. 2 | Differences in the fecal microbial communities of Finnish and Malawian children at 6, 18, and 30 months of age.** Alpha diversity presented as Chao1 and Simpson diversity (1-D) (left panel). The medians with interquartile ranges are indicated. Differences are analyzed using two-sided Mann–Whitney U test and *p*-values are marked in the graphs. Gut community composition presented as NMDS of Bray–Curtis dissimilarity metrics showing clear separation of gut community profiles between Finland and Malawi and between ages within each country (right panel). Composition differed significantly between countries at every sampling age and between 6 and 30 months of age within both countries. For Finnish children *n* = 40 and for Malawian children *n* = 40 biologically independent samples.

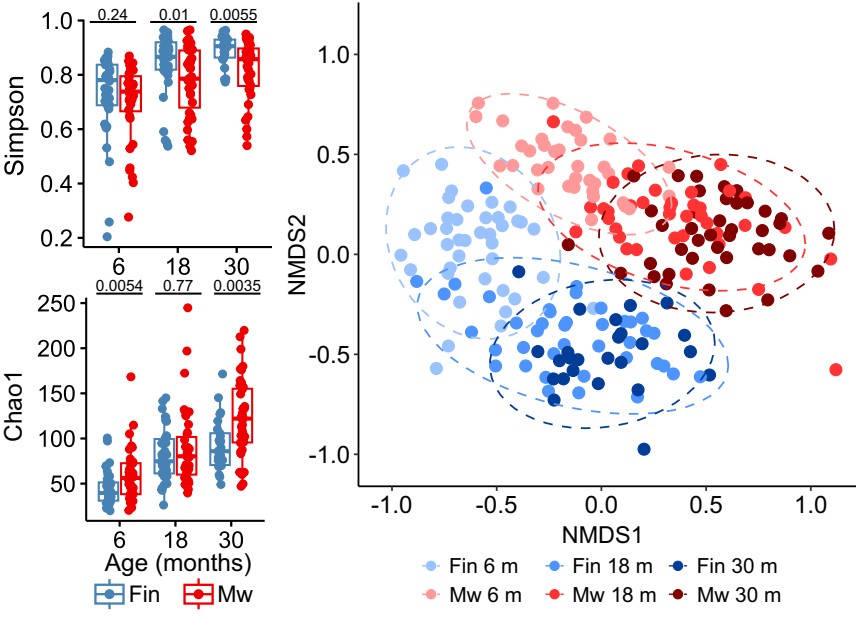

during this period (microbiota might harbor different species at different ages).

In Finnish children, a *Campylobacter* operational taxonomic unit (OTU) was associated with higher IL-10 plasma concentrations at 6 months of age, whereas OTUs belonging to the genera *Clostridium XVIII* and *Enterococcus* were associated with lower IL-10 levels at 6 months and *Coprobacillus* OTU at 18 months. In the Malawian children, gut microbial alpha diversity (Simpson) was associated directly, and *Staphylococcus, Enterococcus*, and *Ruminococcus2* OTUs were associated inversely with plasma IL-10 (Figs. 4 and 5).

For IL-6, viral associations were seen between norovirus and IL-6 in Malawi at 6 months and enterovirus and IL-6 in Finland at 18 months. In addition, Streptococcus OTU and microbial diversity (PD) were associated with IL-6 in Finnish children at 6 months (Figs. 4 and 5).

One *Fusicatenibacter* OTU was associated with higher IL-1β levels, and *Staphylococcus* and *Clostridium sensu stricto* OTUs were associated with lower IL-1β levels in Finland at 6 months of age. At 18 months, *Coprococcus* OTU was associated with higher IL-1β levels in plasma in Finland, and a positive association between an *Intestinimonas* OTU and IL-1β was found in Malawian children. *Clostridium XIVa* OTU was inversely associated with IL-1β in Malawi at 18 months. (Figs. 4 and 5).

An OTU from the genera *Bilophila* and *Ruminococcus2* was associated with lower TNF-α levels at 6 months in Finland, whereas *Streptococcus* OTU was associated with lower TNF-α concentrations in Malawi at 18 months of age (Figs. 4 and 5).

## Discussion

Environmental variability in microbial exposures, infectious disease burden, diet, and lifestyle, along with genetic factors, are important for modulating immune responses and might result in differences in immune phenotypes between populations. Many areas in Africa are currently witnessing the Westernization of lifestyles and the increasing prevalence of immune-mediated diseases. Our analysis of immunological and microbial markers in the sample series of children living in East Africa (rural Malawi) and Northern Europe (Finland) up to three years of age showed that immunological and microbial differences are seen already in early childhood. The Malawian children had significantly higher plasma concentration of all cytokines measured (IL-10, IL-6, IL-1β, TNF-α) compared to the Finnish children, and the longitudinal trajectories of IL-10, IL-6, and TNF-α differed between the groups during follow-up. This result is in line with previous findings where IL-10 and TNF-α levels in non-stimulated whole blood were

shown to be higher in African (rural and semi-rural areas in Gabonese) than European (Dutch) school-aged children[29] and another work where TNF-α, IL-6, and IL-10 plasma levels were higher in African (Tanzanian) than in European (Dutch) adults[6]. In contrast, Smolen et al.[30] found that in 2-year-old children TNF-α levels were lower in urban South-African children in non-stimulated whole blood compared to urban European (Belgian) children.

Previous studies have suggested that the rural versus urban living environment is associated with differences in immune profiles. Plasma IL-6 and TNF-α levels were shown to be higher in 12–36-month-old children living in a rural compared to an urban environment in the South African AmaXhosa population in a study on atopic dermatitis[31]. In addition, PBMC gene expression profiles were also heavily dependent on the rural versus urban environment of the child, and the environment had a substantially greater effect on the gene expression than having atopic dermatitis[32]. These findings emphasize the importance of environmental exposures on the immune system already in early childhood. Furthermore, the level of immune activation has been reported to follow a gradient from high activation in rural Africans (Senegalese), through urban Africans (Senegalese), toward low activation in Europeans (Dutch) in adults[33]. On the other hand, no difference in immune cell populations between children living in rural and urban environments (with anthroposophic versus non-anthroposophic lifestyle) was found in a European (Swedish) study[34], and only minor differences were observed in a South American (Ecuadorian) study[35]. However, the lack of difference in these studies could be due to the small sample size in the European study (*N* = 15) and smaller environmental differences between the study groups in both studies (e.g., lack of farm animals in the rural group in the European study and ruralized lifestyle of recent rural migrants within the urbanized group in the South American study). Thus, even though differences in host genetic composition are known to influence innate immunity[36], and the genes regulating cytokine responses seem to differ between populations of African and Western European ancestry[37], recent studies have documented substantial effects of environmental factors on most human immune traits[38–41].

Our gut microbiota results are in line with earlier findings. Overall, alpha diversity increased in both countries with increasing age as noted before[9,42]. Higher Chao1 index but lower Simpson index in Malawian samples reflects high richness (especially of rare taxa) and unevenness of taxa in the gut compared to Finnish children. Bacterial community composition (beta diversity) differed at all ages between the countries, and it also changed within the countries along with age. The changes seen within the

**Fig. 3 | Pathogen positivity of Finnish and Malawian stool samples.** Adenovirus, enterovirus, parechovirus, rhinovirus, rotavirus, norovirus, *Cryptosporidium* spp., and *G. lamblia* were detected in stool using RT-PCR at 6, 18, and 30 months of age. Positivity indicated as percent of all samples in each group. Differences between the countries were analyzed with two-sided Fisher's exact test. *$P < 0.05$, **$P < 0.01$, ***$P < 0.001$, ****$P < 0.0001$ between the groups. Exact $P$-values for adenovirus are $P = 0.023$, $P = 0.005$, $P = 0.001$ at 6-, 18- and 30-months-of-age, respectively, enterovirus $P = 3.568e$-12, $P = 5.717e$-14, $P = 8.162e$-11, parechovirus $P = 0.0002$, $P = 0.025$, $P = 0.137$, rhinovirus $P = 0.014$, $P = 0.263$, $P = 0.021$ rotavirus $P = 1$, $P = 0.494$, $P = 1$, norovirus $P = 0.018$, $P = 0.055$, $P = 1$, *Cryptosporidium* spp. $P = 0.017$, $P = 0.12$, $P = 0.279$, *Giardia lamblia* $P = 0.207$, $P = 6.961e$-08, $P = 5.165e$-12.

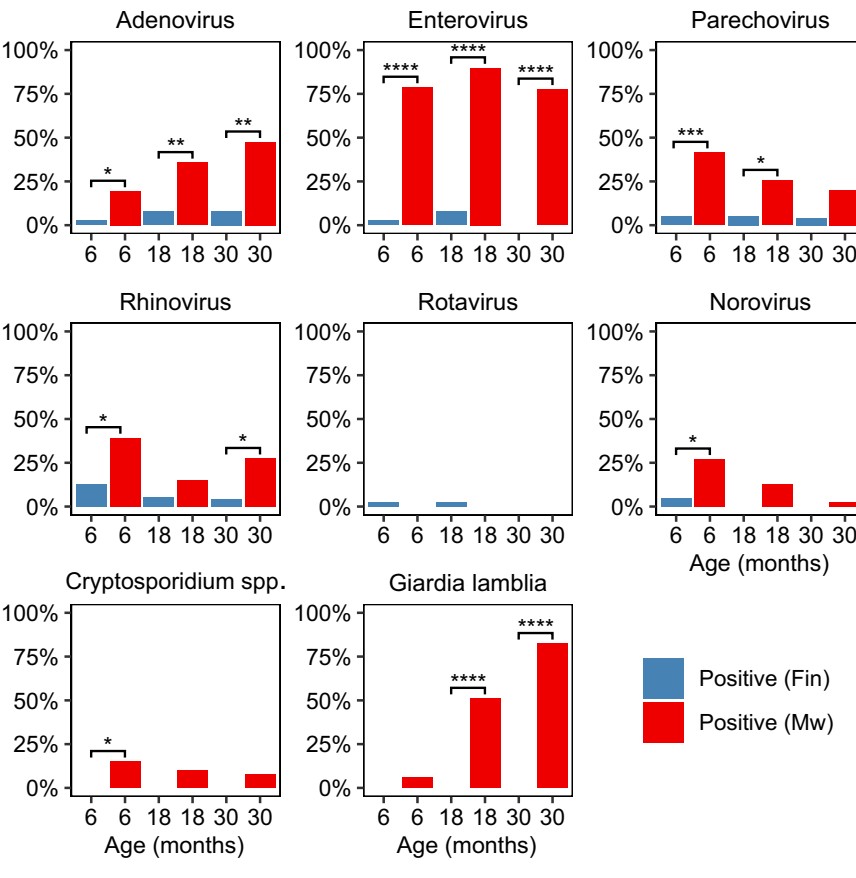

age gradient are connected to the maturation of the microbiota with distinct changes related to introduction of solid foods and weaning during early childhood[9,43–45].

We found that the gut microbiota of the Malawian children matured into a *Prevotella*-dominating type along with age, whereas *Bacteroides* became the dominating genus in the microbiota of the Finnish children. The diet of the Malawian children consisted mostly of carbohydrates (maize porridge, vegetables and fruits) and was very scarce in protein sources representing a diet typically connected to *Prevotella*-dominated gut microbiota (common in rural Africa), whereas the diet of the Finnish children represented typical characteristics of an industrialized population with high levels of fat and protein connected to *Bacteroides*-dominated gut microbiota (Supplemental Table 4)[46–48]. Similar to the immune response, there are some indications that also the gut microbiota would be in a continuum of urban-rural and geographical gradient ordered from rural Africans, through urban Africans towards Europeans[49,50].

According to our results, the pathogenic burden of rural Malawians is strikingly heavier compared to Finns already in early childhood, which might be connected to the difference in hygiene conditions, such as the source of drinking water and sanitary facilities (Table 1). Similar data of hygiene conditions was not collected in the Finnish study, but approximately 90% of households in Finland have been connected to a municipal water distribution network, including both piped drinking water and wastewater systems[51]. The rest usually have a borehole as a source of drinking water. The pathogen positivity in Malawian samples was similar to earlier studies[52,53]. Adenovirus, enterovirus, parechovirus, rhinovirus, norovirus, *Cryptosporidium* spp., and *G. lamblia* were all more common in the stool of Malawian children at some or all ages studied. In addition, many of the Malawian children had multiple pathogen positivity in the same sample. The immune system in Malawian children is continuously challenged by exposure to different types of microbes starting early in life, which might lead to a state of heightened immune activation[54] and earlier T cell maturity

in children[55], making the immune system more unresponsive to individual microbial exposures[56]. This unresponsiveness might explain why we found substantially fewer associations between systemic cytokine levels and single microbes in the Malawian children than in the Finnish children. In addition, other infections not detected in this analysis might mask the associations between the studied pathogens and immunomarkers.

The machine learning feature selection method revealed several potential associations with the circulating levels of cytokines at 6 and 18 months of age. A *Campylobacteria* OTU was associated with higher IL-10 plasma levels in Finland at 6 months. *Campylobacter jejuni* has been shown to induce IL-10 expression in human cells[57,58]. IL-10 is thought to control inflammatory response to *C. jejuni*, as knockout mice lacking IL-10 develop severe enterocolitis after *C. jejuni* infection[59]. On the other hand, *Campylobacteria* may benefit from IL-10 production, as this might allow the bacteria to persist longer. In chickens, breeds that produce higher levels of IL-10 following *Campylobacter jejuni* colonization act as asymptomatic carriers compared to breeds with low IL-10 production[60].

An *Enterococcus* OTU was inversely associated with IL-10 in both Finnish and Malawian children at 6 months. Constant exposure to high levels of this *Enterococcus faecium* has been shown to reduce expression of IL-10 in piglets after weaning[61].

An OTU from the short-chain fatty acid (SCFA) -producing genus *Ruminococcus* group 2[62] was associated with lower TNF-a levels in Finnish children, whereas another OTU from the same genus was associated with lower IL-10 levels in Malawian children. Reduction of TNF-α production by SCFAs has been reported before[63], but genus *Ruminococcus* 2 has also been linked to lower IL-10-producing T regulatory cell numbers in rheumatoid arthritis patients[64].

*Staphylococcus* OTU was inversely associated with IL-1β at 6 months of age in Finnish children. *Staphylococcae* are the most prominent bacteria in human milk[65]. In this study, breastfed children had higher IL-10, which might downregulate proinflammatory IL-1β expression and explain the

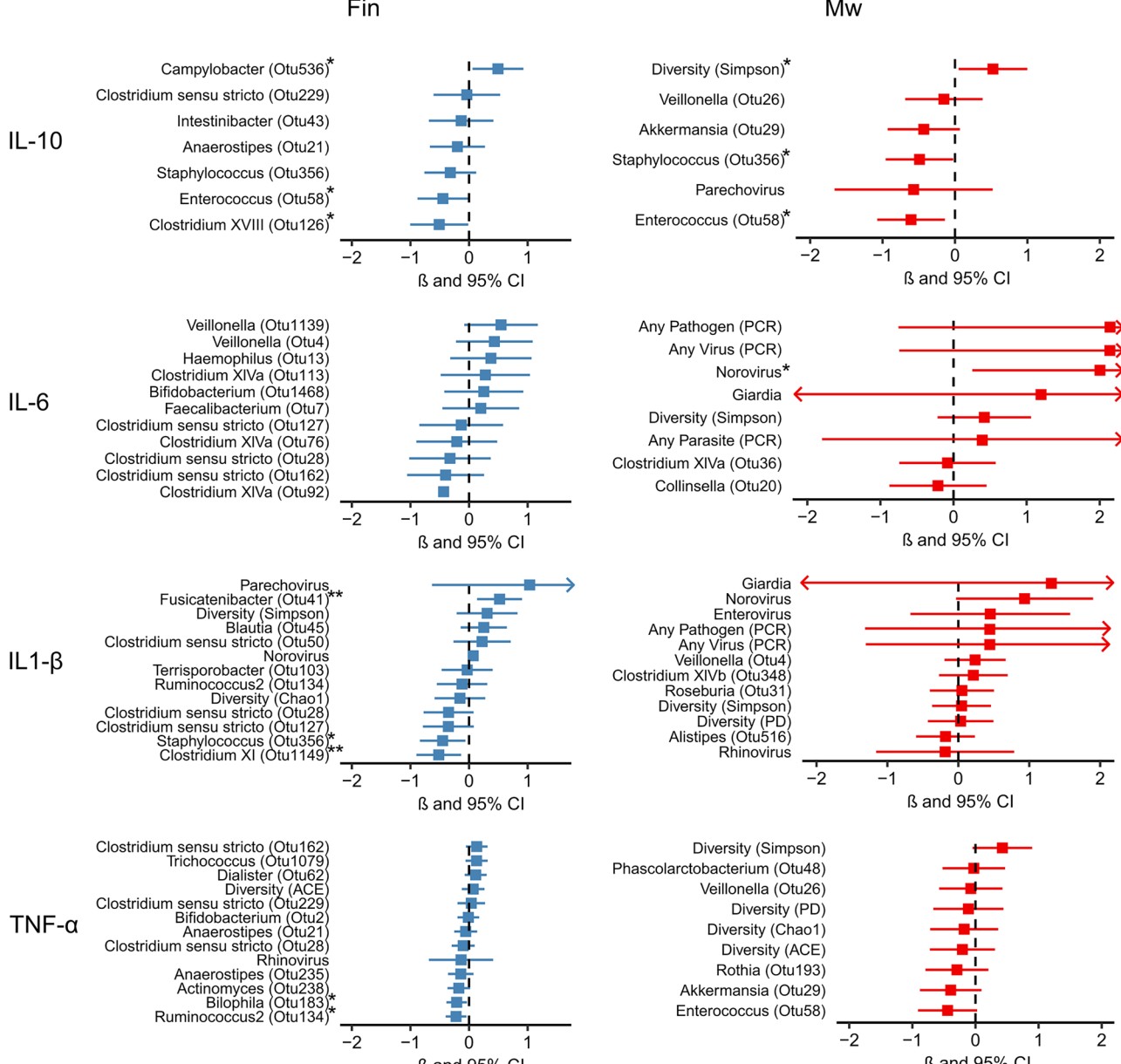

**Fig. 4 | Most important gut microbial features for each plasma cytokine in Finnish and Malawian children at 6 months of age.** Random forest and elastic net regularization selected gut microbial features associated with plasma cytokine levels at 6 months of age for IL-10, IL-6, IL-1β, and TNF-α. The linear regressions were adjusted for sex, weight-for-age Z-score, and breastfeeding. Beta (β) coefficient represented as a square and 95% confidence interval as a line. Arrow indicates a confidence interval continuing beyond the x-axis. *P < 0.05, **P < 0.01 of two-sided student's t test for associations without adjustments for multiple comparisons. Exact

P-values for the statistically significant models of Finnish children: *Campylobacter* OTU P = 0.027, *Enterococcus* OTU P = 0.013, *Clostridium XVIII* OTU P = 0.044, *Fusicatenibacter* OTU P = 0.009, *Staphylococcus* OTU P = 0.040, *Clostridium XI* P = 0.009, *Bilophila* P = 0.019, and *Ruminococcus 2* OTU P = 0.016. Exact P-values for the statistically significant models of Malawian children: Simpson P = 0.032, *Staphylococcus* OTU P = 0.040, *Enterococcus* OTU P = 0.013, and Norovirus P = 0.019.

inverse association with *Staphylococcus* OTU. Altogether, the results suggest that the dynamic changes in gut microbiota and acute intestinal infections can have a marked effect on the immune system during the first months of life.

To our knowledge, this is the first longitudinal study comparing the immunological and microbial markers in rural African and European children during early childhood. Although immune marker and microbiota comparisons between the populations have been done earlier[6,29,50] differences during early childhood have not been reported before. In addition, previous immune marker studies often include only one sampling age in contrast to longitudinal follow-up, which provides a more complete picture of the differences.

It is also important to consider the limitations of the current study when interpreting the results. First, a larger sample size would have facilitated a more robust analysis of associations between microbial exposures and plasma cytokine levels. Second, it is challenging to completely exclude all confounding factors in this kind of comparative study. The sample collection methods and period (1999–2012 in Finland and 2012–2015 in Malawi) differed between the groups which might affect the cytokine detection in plasma samples and microbiome results of the stool samples. However, the blood and stool samples were stored at -80 °C in both groups. Also, the sample collection was performed similarly although not identically and the differences seen in the plasma cytokine concentrations between the two groups are in line with previous observations which support the

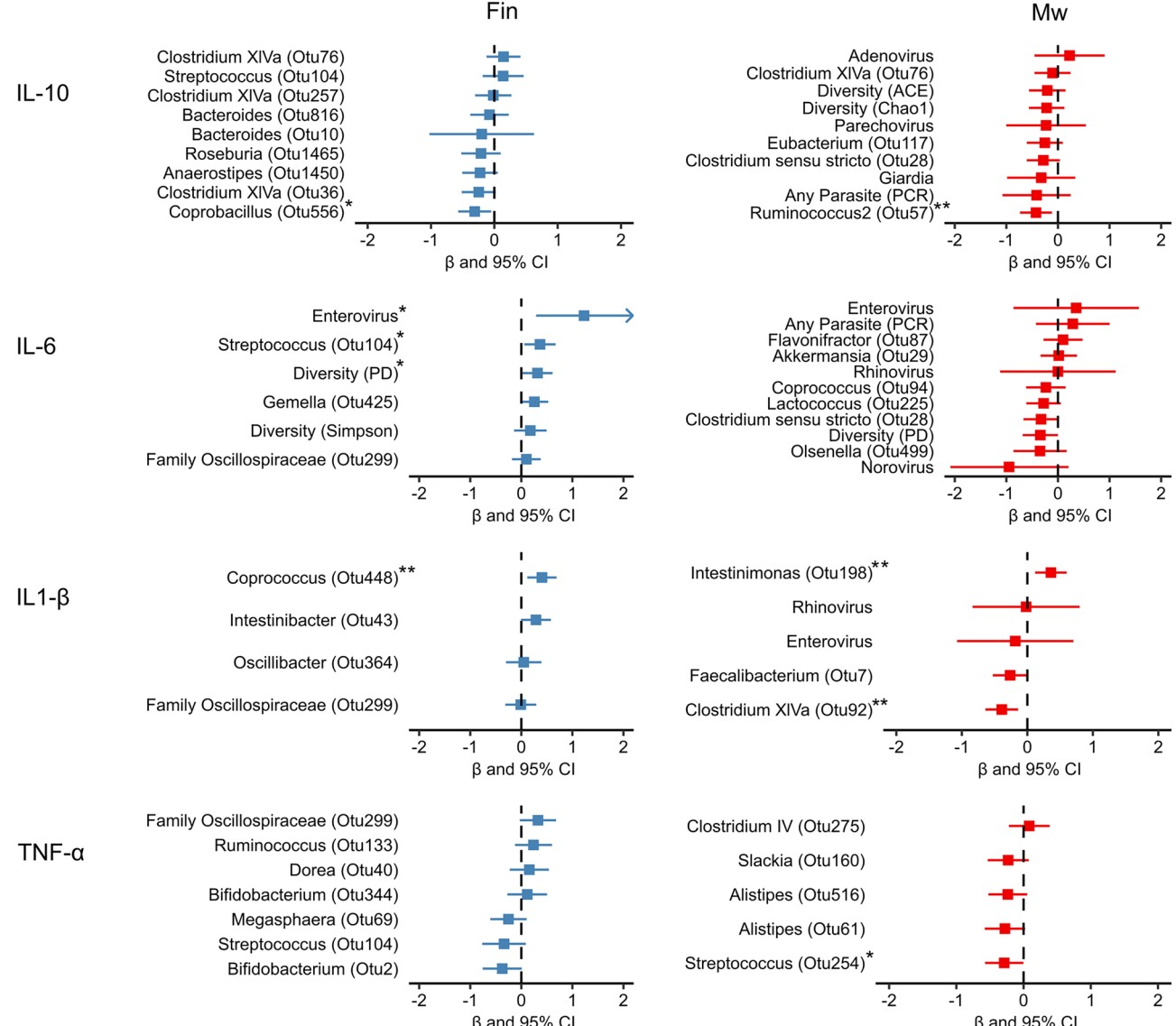

**Fig. 5 | Most important gut microbial features for each plasma cytokine in Finnish and Malawian children at 18 months of age.** Random forest and elastic net regularization selected gut microbial features associated with plasma cytokine levels at 18 months of age for IL-10, IL-6, IL-1β, and TNF-α. The linear regressions were adjusted for sex, weight-for-age Z-score, and breastfeeding. Beta (β) coefficient represented as a square and 95% confidence interval as a line. Arrow indicates a confidence interval continuing beyond the x-axis. *P < 0.05, **P < 0.01 of two-sided student's t test for associations without adjustments for multiple comparisons. Exact P-values for the statistically significant models of Finnish children: *Coprobacillus* OTU P = 0.021, enterovirus P = 0.016, *Streptococcus* OTU P = 0.021, PD P = 0.036, *Coprococcus* OTU P = 0.007. Exact P-values for the statistically significant models of Malawian children: *Ruminococcus 2* OTU P = 0.009, *Intestinimonas* OTU P = 0.005, *Clostridium XIVa* OTU P = 0.003, *Streptococcus* OTU P = 0.049.

genuineness of the observed differences. Furthermore, the difference between TNF-α, IL-1β and IL-10 does not seem to be explained by the difference in storage time (Supplementary Fig. S6) and also IL-6 has been shown to be stable at −80 °C[66]. In addition, seasonality does not seem to explain the differences in the cytokines since all seasons are represented in both groups (Supplementary Fig. S7). Also, nucleic acids in the stool are protected by viral capsid or parasitic oocyte and are therefore markedly more resistant than naked nucleic acids. Our previous studies have shown that detection of viruses by PCR in stools is not affected by overnight exposure to ambient temperatures[67]. Therefore, it is unlikely that the marked difference between the countries could be explained by degradation of viral RNA during sampling or storage. Third, the Finnish participants in the study were selected based on HLA screening thus they do not represent the general population. Since HLA genotypes can modulate immune responses to microbes[68], we cannot exclude the possibility that HLA-related factors could also have contributed to the observed differences in blood cytokines

between the two populations. The relatively small size of the current cohort limited our ability to study such complex interactions between different microbes and different HLA types in these populations. Finally, the detection of rotaviruses might have been affected by the difference in the timing of rotavirus immunization programs. Public rotavirus vaccinations were initiated in 2009 in Finland and in 2012 in Malawi. Thus, most of the Malawian samples are collected during the ongoing rotavirus vaccination program, whereas most of the Finnish samples are collected before the vaccinations started. The rate of rotavirus in stools was also low, which is in line with previous studies[52,69,70]. Low detection rate of rotavirus could be linked to its rapid disappearance after the acute phase of infection.

To summarize, there was a clear difference in plasma cytokine levels, gut microbiota, and pathogen positivity in early childhood in Finland and rural Malawi. The results support the hypothesis that frequent occurrence of infections and higher bacterial diversity could influence the balance of the immune system. This emphasizes the need for further studies exploring

microbiological and immunological effects of the ongoing lifestyle transition in Africa and its possible contribution to the burden of immune-mediated diseases in the future.

## Data availability

The bacteriome sequencing data generated in this study have been deposited in the Sequence Read Archive at NCBI under accession number PRJNA1183592. The source data for the main figures are available in Supplementary Data 1.

## Code availability

Statistical analysis was done using the R language version 4.1.0. using packages phyloseq v1.36.0, vegan v2.6-4, ggplot2 v3.5.1, randomForest 4.6-14, glmnet 4.1-1, dplyr 1.0.6, lme4 1.1-36. Software programs used to analyze data are freely available. All other data relevant to the study are included in the article. Additional data are available on request.

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

## Acknowledgements

The authors wish to thank the participants, their families, and the study staff of DIPP and iLiNS-DYAD-M studies. Tanja Kuusela, Minta Lumme, and Sari Valorinta (Faculty of Medicine and Health Technology, Tampere University) are acknowledged for their technical support. This work was supported by Business Finland (grant numbers 40333/14 and 6766/31/2017 ADELE project) and Tampere Tuberculosis Foundation. NN has received personal grants from Pirkanmaa Regional fund of Finnish Cultural Foundation, Juho Vainio Foundation, Child and Nature Foundation, and Orion Research Foundation sr. The DIPP study is supported by JDRF (grants 1-SRA-2016-342-M-R, 1-SRA-2019-732-M-B, 3-SRA-2020-955-S-B); Novo Nordisk Foundation; Academy of Finland (Decision No 292538 and Centre of Excellence in Molecular Systems Immunology and Physiology Research 2012-2017, Decision No. 250114); Special Research Funds for University Hospitals in Finland; The Foundation for Pediatric Research, Helsinki, Finland; Sigrid Juselius Foundation, Finland and the Diabetes Research Foundation, Finland.

## Author contributions

N.N. drafted the manuscript and contributed to the conceptualization, analysis, and interpretation of data. Y-M.F., E.K., L.H., K.M., and K.M.L. contributed to the acquisition of the data. J.L. contributed to the analysis and interpretation of data, O.H.L. and A.S. contributed to the conceptualization of the work, Jo.L., J.T., R.V., M.K., and K.K. contributed to the acquisition of the data. H.H., P.A., S.O., and U.A. contributed to the conceptualization and design of the work and the acquisition of the data. All authors revised the manuscript critically for important intellectual content and gave final approval of the version to be published.

## Competing interests

A.S., H.H., O.H.L., N.N., and S.O. have been named as inventors in a patent application "immunomodulatory compositions" submitted by the University of Helsinki (Patent application number 20165932 at Finnish Patent and Registration Office). R.V. has been a member of the Sanofi Advisory Board since 2023. All other authors declare no competing interests.

## Additional information

[1]Faculty of Medicine and Health Technology, Tampere University, Tampere, Finland. [2]Faculty of Medicine, University of Helsinki, Helsinki, Finland. [3]School of Global and Public Health, Kamuzu University of Health Sciences, Blantyre, Malawi. [4]Faculty of Veterinary Medicine, University of Helsinki, Helsinki, Finland. [5]Luke, Natural Resources Institute Finland, Turku, Finland. [6]Immunogenetics Laboratory, Institute of Biomedicine, University of Turku, Turku, Finland. [7]Department of Pediatrics, Turku University Hospital, University of Turku, Turku, Finland. [8]Clinical Microbiology, Turku University Hospital, Turku, Finland. [9]Centre for Population Health Research, University of Turku and Turku University Hospital, Turku, Finland. [10]Research Centre for Integrative Physiology and Pharmacology, Institute of Biomedicine, University of Turku, Turku, Finland. [11]InFLAMES Research Flagship Centre, University of Turku and Åbo Akademi University, Turku, Finland. [12]Department of Pediatrics, Research Unit of Clinical Medicine, Medical Research Centre, Oulu University Hospital, University of Oulu, Oulu, Finland. [13]Department of Pediatrics, Tampere University Hospital, Tampere, Finland. [14]Seinäjoki University Consortium, Seinäjoki, Finland. [15]Research Program for Clinical and Molecular Metabolism, Faculty of Medicine, University of Helsinki, Helsinki, Finland. [16]Fimlab Laboratories, Tampere, Finland. ✉e-mail: heikki.hyoty@tuni.fi

