## [Transparent Peer Review file · Communications Medicine]

Early-life immunological and microbial differences between East African and North European children

Corresponding Author: Professor Heikki Hyoty

Version 0:

Reviewer comments:

Reviewer #1

(Remarks to the Author)

Brief Summary of the Manuscript

This study compares immune, microbiome, and pathogen positivity in early childhood between Malawian and Finnish children. The authors analyze plasma cytokine levels (IL-10, IL-6, IL-1 β , TNF- α), gut microbiota diversity, and pathogen presence at multiple time points (6, 18, 24 and 30 months). They find that Malawian children exhibit higher immune activation, greater gut microbial diversity, and increased pathogen carriage than Finnish children. These findings support the hypothesis that the current wave of urbanization in many communities in Africa contributes to the differences in microbial exposure thereby affecting immune system development and may contribute to the increasing incidence of immune-mediated diseases in urbanizing African populations.

Overall Impression of the Work

The study addresses an important global health issue—how environmental and microbial exposures in early childhood shape immune function and susceptibility to immune-mediated diseases which are sharply increasing in most of the African populations. The longitudinal design and use of multi-omics approaches (cytokine profiling and gut microbiota analysis) strengthen the study's conclusions. However, there are some areas where clarity, methodological details, and additional analyses could improve the manuscript.

Specific Comments and Recommendations

1. Scientific and Technical Soundness

Strengths:

- The study design includes longitudinal sampling, allowing for the tracking of immune and microbial changes over time.
- The use of plasma cytokine measurements, gut microbiota profiling, and pathogen detection provides a comprehensive dataset.
- The application of statistical modeling (random forest and elastic net regularization) to identify microbial associations with cytokines is robust.

Areas for Improvement & Recommendations:

- While the study presents strong observational data, there are some gaps in methodology, particularly in sample processing and confounding factors that may affect immune responses. The authors should explicitly describe how plasma and stool samples were collected, stored, and analyzed to ensure reproducibility. They should also state whether any potential degradation effects were assessed. Again since the samples in Malawi and Finland were collected at different times the period of storage is also different was this accounted for during analysis?
- Connected to the above comment samples since samples were collected at different time points and previous studies have shown seasonality affects the immune function (see these articles [https://www.cell.com/fulltext/S0092-8674\(16\)31401-5](https://www.cell.com/fulltext/S0092-8674(16)31401-5), <https://www.nature.com/articles/s41590-021-00867-8>)
- Authors could consider adding in line 191 the following reference which also details the genetic regulation of cytokine response comparing African and European populations [https://www.cell.com/ajhg/fulltext/S0002-9297\(22\)00014-3](https://www.cell.com/ajhg/fulltext/S0002-9297(22)00014-3)
- While the study accounts for some variables (e.g., stunting, breastfeeding), other factors such as diet, hygiene practices, or antibiotic use could influence both immune responses and gut microbiota composition additional analysis accounting for these parameters could enrich the manuscript if this data were not collected authors should consider an additional discussion on these confounders would improve the conclusions.

2. Clarity of Result Reporting

Strengths:

Figures and supplementary materials are well-integrated into the results section.

-Key findings (differences in cytokine levels, microbial diversity, and pathogen positivity) are clearly described.

Areas for Improvement & Recommendations:

a) Comparability of Sampling Time Points: There is a mismatch in sampling time points between the two cohorts (Malawian children at 30 months vs. Finnish children at 24 and 36 months). The authors compare these groups but should justify why this approach is valid. Consider an additional analysis accounting for the time point differences and how this might have affected the findings.

3. Methodological Clarity

a) The methodology does not provide sufficient detail on sample collection, storage, and processing. Given that cytokines and microbial profiles are sensitive to sample handling, more details are needed. The authors should explicitly describe how plasma and stool samples were collected, transported was it in a cold chain or RT, Separation at what RPM and time, and analyzed to ensure reproducibility. They should also state whether any potential degradation effects were assessed.

b) In the method section eg cytokine measurement a briefly detail on how this assay was performed will aid readability and improve reproducibility. The authors should provide more details in the method section for the different assays performed.

4. Comparisons and Potential Biases

a) The study presents microbial diversity differences but does not account for batch effects. The authors should discuss potential batch effects.

b) The choice of random forest and elastic net models for microbial association analysis is justified, but the study lacks validation. The authors should include cross-validation details and discuss the robustness of their models.

c) The study focuses on microbial and cytokine differences but does not explore host metabolic responses. The authors should consider metabolomics to understand whether microbial differences translate into functional metabolic changes that influence immune development.

Reviewer #2

(Remarks to the Author)

Authors present a descriptive study which aims to link the microbiota with inflammatory status in two geographically distinct populations, with potential relevance to immune mediated diseases. Overall the data are clearly presented and statistical analyses seem appropriate. A major advantage of this study is the longitudinal nature of the samples collected – though majority of analyses seem to be limited to comparing specific timepoints, it may be interesting to see if time can be integrated into more analyses (e.g. cytokine data). The authors also used machine learning techniques to identify components of the stool microbiome associated with cytokine levels, and different taxa were identified in each cohort, however discussion of these findings was lacking. Was this expected? Have these bacteria previously been associated with innate immune markers or immune mediated diseases in other studies?

Findings of the study are repeatedly discussed in the context of the higher prevalence of immune mediated diseases in European societies compared to rural African societies, though no part of the study reports immune mediated disease incidence in the cohorts. Non-diabetic European infants were specifically selected for analysis, which were from a larger Finnish study assessing T1D; though it is not clear if similar exclusions were applied to the Malawi study. If the study aims to identify differences in microbiota that could explain different propensities for autoimmune development, would the inclusion of some autoimmune participants not strengthen the work?

The issue of Westernisation of African communities is frequently discussed but it is not clear if the data presented in the Malawi cohort are intended to represent “pre-“ or “post-“ westernisation samples - are the differences reported lower than those historically reported? Are microbiome differences between these populations getting smaller over time?

As conclusions of this study rely on comparing data generated from biosamples from two different cohorts in separate locations over different periods of time, there is a reasonable potential for the data reported here to be biased by biosample collection, storage and analysis, which should be acknowledged and discussed (e.g. heparin vs citrate anticoagulant used, storage conditions of plasma, storage time prior to analysis; See PMID: 19785746) – especially given that cytokine concentrations appear to be globally decreased in Finnish samples. Would any immune regulatory cytokines be expected to be increased?

Viral quantification in all the Malawi samples was consistently higher than in the Finnish samples, could differences in storage have contributed to this? (e.g. Finnish stool samples were shipped overnight before freezing, while Malawi samples were frozen within 6h as reported in PMID: 31436705). However, the cytokine and pathogen positivity findings presented here are generally in agreement with prior reports, which suggests that these differences are not primarily caused by differences in biospecimen collection.

More specific comments:

Abstract line 52: 'if' should be 'of'

Figure 2 lacks labels (A and B)

OTU is not defined in the text

Page 7 line 191 'have documented a substantial effects': the 'a' can be removed if discussing multiple effects

Page 8 Line 218: 'less associations' should be 'fewer associations'

Text in Figure S2 is too pixelated to read.

Reviewer #3

(Remarks to the Author)

The study "Early-life immunological and microbial differences between East African and North European children – implications for the rising prevalence of immune-2 mediated diseases in Africa" explores differences in gut microbiome composition between urban Finnish population and rural Malawian population of infants and their relations to the levels of cytokines. The authors present novel data as well as statistical analysis. The data analysis includes statistical tests on the group differences and measures of feature importances using three different methods, applied sequentially (Random Forest, linear regression with elastic net regularization, linear regression).

While the study investigates an important issue of prevalence of immune-modulated diseases in relation to lifestyle, the machine learning methods are currently insufficiently described. The reviewer poses the following questions, answering which could help the paper's clarity.

Major comments:

1. It says in the methods that all the features were "normalized", however the authors do not describe the normalization method. Were the compositional microbial data log-transformed? Using compositional features as raw counts or percentages in a linear regression violates the independence assumption, making the interpretation harder. The reviewer therefore proposes that the authors report the data transformations in detail.
2. All three applied methods (Random Forest, linear regression with elastic net regularization, linear regression) allow the selection of the most important features. Why did the authors consider it necessary to apply Random Forest before the linear regressions? Do the slope coefficients change if the regression is done on all features, without pre-filtering? Similarly, why couldn't the slope coefficients for elastic net regression be used for the final conclusions, without an extra linear regression? The reviewer proposes performing the analysis with one linear regression with elastic net regularisation, without the Random Forest and an extra linear regression step.
3. The full model specification for the resulting linear regression must be included, with the list of all features and the corresponding beta-coefficients.
4. It would help the consistency if the length-for-age Z-score is used in the regression instead of weight-for-age Z-score, since it's the measure that is used for defining stuntedness.
5. If the authors demonstrate the utility of the Random Forest algorithm, its arguments must also be reported.
6. Please provide access to the R code in a public code repository.

Minor comments:

1. Line 52: a minor typo "if" → "of"
2. It's very hard to see the microbial composition from the supplementary figure S2. It would be very helpful if it could be provided as a separate table.

Version 1:

Reviewer comments:

Reviewer #1

(Remarks to the Author)

Reviewer #1 Final Statement

The authors have carefully and comprehensively addressed all of my previous comments. The additional methodological details, clarifications, and supplementary analyses provided have significantly improved the scientific soundness, reproducibility, and overall clarity of the manuscript. I also appreciate the inclusion of the suggested reference and the expanded discussion of potential confounders such as diet, hygiene, and seasonality. The revision substantially strengthens the study and its conclusions. I commend the authors for their thorough and thoughtful responses and have no further comments.

Reviewer #2

(Remarks to the Author)

I thank the authors for attempting to address the review comments. Concerns I raised regarding descriptions of biospecimen handling, and the clarity of Fig. S2 have been addressed well. However, I have two concerns for the revised version of this manuscript:

The added discussion about microbe-cytokine associations is severely lacking. The link between campylobacter and IL10 is well appreciated (IL10 knockouts are a standard mouse model for symptomatic c jejuni infection), which has not been well communicated in the text. The following paragraph discusses literature on Intestinibacter, which is distinct from Intestinimonas and Clostridium, and so it is not clear why this is relevant. In my view, a major strength of a study with this design (microbiome and pathogen load/immune parameters assessed in the same individuals) is the ability to find these associations - contextualizing these should be a considerable focus of the discussion.

I also find that the study has little relation to immune-mediated diseases as these are not included in any of the analyses, nor any specific biomarkers of these (e.g. autoantibodies), and any links to immune related diseases appear speculative. The Finnish cohort also appears to exclude children with HLAs associated with T1D, which will create bias in the data. Justification that HLA types do not impact cytokine levels is incorrect, as multiple studies have shown these to be linked, and also with inflammatory diseases (PMID: 39697331, PMID: 40572779, PMID: 32127039). The findings the study has made are not appropriately contextualized with knowledge on the etiology of these diseases. Thus, I do not believe this study contributes any specific finding related to immune-related diseases - it is in my view inappropriate to reference immune-related diseases in the title of this manuscript.

The introduction states: "Despite this possible connection little research is done on early-life immune status, microbiota and microbial infections" - I do not believe this is the case as there is a strong body of literature linking these ideas e.g. PMID: 40482668, PMID: 35550670, PMID: 40175554, PMID: 37138015, PMID: 31964813, PMID: 31719945, PMID: 29602225, PMID: 24637604, PMID: 36419421, PMID: 37764797 etc

Reviewer #3

(Remarks to the Author)

The authors presented a longitudinal study comparing the trajectories of microbiome development of an African and a Western population. The reviewer's concerns were regarding the computational and statistical methods and were clarified by the authors in the changes that were made in the main text and the Methods section. The reviewer thus does not have follow up comments.

Version 2:

Reviewer comments:

Reviewer #2

(Remarks to the Author)

The authors have made substantial efforts to address concerns raised in previous reviews - including acknowledging limitations of the study, removing the titular focus on immune-mediated diseases and improving discussion of findings within the current literature. In general, my view is that discussion of immune-mediated diseases is still a little heavy given lack of data on this aspect. Additionally, a major advantage of this study appears to be the longitudinal sampling, however findings related to this aren't really novel (e.g. its already known that the microbiota alpha diversity changes across these timepoints). Earlier I suggested including changes over time in the analyses between the populations - i.e. are there differences between the populations in how the microbiota or innate immune landscape change over time? - which I believe would strengthen the paper. The introduction/abstract mentions that 'characteristics of the immune system' were measured, which sounds more comprehensive than what was actually measured (four innate immune cytokines). e.g. "the characteristics of the immune system" - perhaps just change to "cytokines" or "innate immune cytokines"?

My additional suggestions are minor:

Some editing is needed across the manuscript to avoid awkward-sounding phrasing - e.g. "Many diseases linked to immune system" should read "Many diseases linked to the immune system"; 'lifestyles' is more appropriate than 'lifestyle' in many cases.

Two consecutive sentences appear in the introduction that begin with "Using longitudinal analysis and machine learning, the current study aims to explore overall and temporal differences in..."

Responses to reviewers' comments

Reviewer #1 (Remarks to the Author):

1. Scientific and Technical Soundness

Areas for Improvement & Recommendations:

a) While the study presents strong observational data, there are some gaps in methodology, particularly in sample processing and confounding factors that may affect immune responses. The authors should explicitly describe how plasma and stool samples were collected, stored, and analyzed to ensure reproducibility. They should also state whether any potential degradation effects were assessed. Again since the samples in Malawi and Finland were collected at different times the period of storage is also different was this accounted for during analysis?

Authors' response: We have now added more detailed description of sample collection and storage in the Methods section for the Malawian (page 11) and Finnish (page 13) samples. We also added graphs of the Finnish plasma cytokine levels against sampling date in Supplementary Figure S5 to visualize possible effect of storage time and discussed this data on page 10. TNF-a, IL-1b ja IL10 differences do not seem to be explained by the length of storage. IL-6 could be affected by storage time, however, earlier studies has shown the stability of IL-6 at -80 °C (<https://doi.org/10.1016/j.cyto.2020.155057>).

b) Connected to the above comment samples since samples were collected at different time points and previous studies have shown seasonality affects the immune function(see these articles [https://www.cell.com/fulltext/S0092-8674\(16\)31401-5](https://www.cell.com/fulltext/S0092-8674(16)31401-5), <https://www.nature.com/articles/s41590-021-00867-8>

Authors' response: It is true that seasonality can affect immune function. There are three seasons in Malawi located in the tropics May-August: cool and dry, September-October: warm and dry, and November-April: warm and wet. At the same time, there are four seasons in Finland located in the northern hemisphere March-May: spring, June-August: summer, September-November: autumn, and December-February: winter. The compared study sites have different meteorological seasons and their effect on infections and other factors modifying cytokine concentrations in blood is so different that it is challenging to compare seasonal effects between the groups. For this study it is important that the samples cover all seasons. We added Supplementary Figure S6 to present the concentrations of plasma cytokines in each season in both groups and the distribution of blood sampling season within each sampling age.

c) Authors could consider adding in line 191 the following reference which also details the genetic regulation of cytokine response comparing African and European populations [https://www.cell.com/ajhg/fulltext/S0002-9297\(22\)00014-3](https://www.cell.com/ajhg/fulltext/S0002-9297(22)00014-3)

Authors' response: We have now included the suggested reference in the manuscript.

d) While the study accounts for some variables (e.g., stunting, breastfeeding), other factors

such as diet, hygiene practices, or antibiotic use could influence both immune responses and gut microbiota composition additional analysis accounting for these parameters could enrich the manuscript if this data were not collected authors should consider an additional discussion on these confounders would improve the conclusions.

Authors' response:

We have now added data on diet in Supplementary Table 2 and discussed the topic on page 8. We also added data on hygiene practices (source of drinking water ja sanitary facility) from Malawian study in Table 1 and data from Finland in Discussion section (page 8).

2. Clarity of Result Reporting

Areas for Improvement & Recommendations:

a) Comparability of Sampling Time Points: There is a mismatch in sampling time points between the two cohorts (Malawian children at 30 months vs. Finnish children at 24 and 36 months). The authors compare these groups but should justify why this approach is valid. Consider an additional analysis accounting for the time point differences and how this might have affected the findings.

Authors' response: We modified Supplementary Figure S1 to allow easier comparison between each time point and to better show that there are no statistical differences between Finnish 24-month and 36-month samples. We also added this information in the Results section (page 3)

3. Methodological Clarity

a) The methodology does not provide sufficient detail on sample collection, storage, and processing. Given that cytokines and microbial profiles are sensitive to sample handling, more details are needed. The authors should explicitly describe how plasma and stool samples were collected, transported was it in a cold chain or RT, Separation at what RPM and time, and analyzed to ensure reproducibility. They should also state whether any potential degradation effects were assessed.

Authors' response: We have now added more detailed description of sample collection and storage in the Methods section for Malawian (page 11) and Finnish (page 13) samples.

b) In the method section eg cytokine measurement a briefly detail on how this assay was performed will aid readability and improve reproducibility. The authors should provide more details in the method section for the different assays performed.

Authors' response: We have now added more details in the description of different analyses in the Methods section (page 14-15).

4. Comparisons and Potential Biases

a) The study presents microbial diversity differences but does not account for batch effects. The authors should discuss potential batch effects.

Authors' response: We performed the PERMANOVA analyses again with NGS batch number as a covariate and the results did not change. We added a sentence about this in the Results section (page 4). We also added batch number as a covariate in the linear regression models.

b) The choice of random forest and elastic net models for microbial association analysis is justified, but the study lacks validation. The authors should include cross-validation details and discuss the robustness of their models.

Authors' response: In random forests, there is no need for cross-validation or a separate test set to get an unbiased estimate of the test set error. It is estimated internally during the run. More information can be found from the webpages of the randomForest R package developers

(https://www.stat.berkeley.edu/~breiman/RandomForests/cc_home.htm#ooberr)

c) The study focuses on microbial and cytokine differences but does not explore host metabolic responses. The authors should consider metabolomics to understand whether microbial differences translate into functional metabolic changes that influence immune development.

Authors' response: The metabolomics data would have helped to get a more complete view on the microbial differences between the groups. Unfortunately, we did not have enough sample volume to perform metabolomics analysis in addition to the microbial analyses.

Reviewer #2 (Remarks to the Author):

Authors present a descriptive study which aims to link the microbiota with inflammatory status in two geographically distinct populations, with potential relevance to immune mediated diseases. Overall the data are clearly presented and statistical analyses seem appropriate. A major advantage of this study is the longitudinal nature of the samples collected – though majority of analyses seem to be limited to comparing specific timepoints, it may be interesting to see if time can be integrated into more analyses (e.g. cytokine data).

Authors' response: We considered doing linear mixed models to analyze cytokine-microbe associations throughout the follow-up time but did not proceed with this strategy since the microbiome of the children is changing so radically and the microbial effects might not be constant during this period.

The authors also used machine learning techniques to identify components of the stool microbiome associated with cytokine levels, and different taxa were identified in each cohort, however discussion of these findings was lacking. Was this expected? Have these bacteria previously been associated with innate immune markers or immune mediated diseases in other studies?

Authors' response: We have now added discussion about the microbe-cytokine associations (page 9).

Findings of the study are repeatedly discussed in the context of the higher prevalence of immune mediated diseases in European societies compared to rural African societies, though no part of the study reports immune mediated disease incidence in the cohorts. Non-diabetic European infants were specifically selected for analysis, which were from a larger Finnish study assessing T1D; though it is not clear if similar exclusions were applied to the Malawi study. If the study aims to identify differences in microbiota that could explain different propensities for autoimmune development, would the inclusion of some autoimmune participants not strengthen the work?

Authors' response: We have now added information about the prevalence of immune-mediated diseases in these countries to the Introduction section (page 2).

The issue of Westernisation of African communities is frequently discussed but it is not clear if the data presented in the Malawi cohort are intended to represent “pre-“ or “post-“ westernisation samples - are the differences reported lower than those historically reported? Are microbiome differences between these populations getting smaller over time?

Authors' response: The Malawian cohort represents rural lifestyle samples (a.k.a “pre-westernization” samples). We have now added more descriptive information about the hygiene conditions in table 1 and discussion section (page 8) in order to make this more clear.

As conclusions of this study rely on comparing data generated from biosamples from two different cohorts in separate locations over different periods of time, there is a reasonable potential for the data reported here to be biased by biosample collection, storage and analysis, which should be acknowledged and discussed (e.g. heparin vs citrate anticoagulant used, storage conditions of plasma, storage time prior to analysis; See PMID: 19785746) – especially given that cytokine concentrations appear to be globally decreased in Finnish samples. Would any immune regulatory cytokines be expected to be increased?

Authors' response: We have now added more detailed description of sample collection and storage in the Methods section for the Malawian (page 11) and Finnish (page 13)

samples. We also added text in the Discussion section about the possible effects of differences in biosample collection and storage (page 10).

Viral quantification in all the Malawi samples was consistently higher than in the Finnish samples, could differences in storage have contributed to this? (e.g. Finnish stool samples were shipped overnight before freezing, while Malawi samples were frozen within 6h as reported in PMID: 31436705). However, the cytokine and pathogen positivity findings presented here are generally in agreement with prior reports, which suggests that these differences are not primarily caused by differences in biospecimen collection.

Authors' response: We have now discussed this on page 10 in the manuscript.

More specific comments:

Abstract line 52: 'if' should be 'of'

Authors' response: We have now corrected this typo

Figure 2 lacks labels (A and B)

Authors' response: We have now corrected the caption of Figure 2

OTU is not defined in the text

Authors' response: We have now added definition of OTU on page 5

Page 7 line 191 'have documented a substantial effects': the 'a' can be removed if discussing multiple effects

Authors' response: We have now removed the extra 'a'

Page 8 Line 218: 'less associations' should be 'fewer associations'

Authors' response: We have now corrected this.

Text in Figure S2 is too pixelated to read.

Authors' response: We have now increased the resolution of Supplementary Figure S2

Reviewer #3 (Remarks to the Author):

Major comments:

1. It says in the methods that all the features were "normalized", however the authors do not describe the normalization method. Were the compositional microbial data log-

transformed? Using compositional features as raw counts or percentages in a linear regression violates the independence assumption, making the interpretation harder. The reviewer therefore proposes that the authors report the data transformations in detail.

Authors' response: The microbial data is now log-transformed for linear regression. The variables were standardized by calculating Z-scores $((x - \mu)/\sigma)$, where μ is the mean value of the variable and σ is the standard deviation of the variable). This information is now added in the Methods section (page 16).

2. All three applied methods (Random Forest, linear regression with elastic net regularization, linear regression) allow the selection of the most important features. Why did the authors consider it necessary to apply Random Forest before the linear regressions? Do the slope coefficients change if the regression is done on all features, without pre-filtering? Similarly, why couldn't the slope coefficients for elastic net regression be used for the final conclusions, without an extra linear regression? The reviewer proposes performing the analysis with one linear regression with elastic net regularisation, without the Random Forest and an extra linear regression step.

Authors' response: Random Forest was done for genus level data whereas elastic net regularization was done for OTU level data. The direction of the slope coefficients did not change when random forest was left out of the analysis pipeline. Linear regression was added after elastic net regularization to include adjustments for sex, weight-for-age Z-score, and breastfeeding and to assess the statistical significance of the models.

3. The full model specification for the resulting linear regression must be included, with the list of all features and the corresponding beta-coefficients.

Authors' response: We have now added Supplementary Data 1 containing full model specification for all linear regression models.

4. It would help the consistency if the length-for-age Z-score is used in the regression instead of weight-for-age Z-score, since it's the measure that is used for defining stuntedness.

Authors' response: We appreciate this suggestion. However, weight-for-age Z-score was found to be more strongly associated with cytokine levels, therefore we decided to include it in the linear regression models instead of length-for-age Z-score.

5. If the authors demonstrate the utility of the Random Forest algorithm, its arguments must also be reported.

Authors' response: We have now added more information in the Methods section (page 16)

6. Please provide access to the R code in a public code repository.

Authors' response: R code is now added as a supplementary file.

Minor comments:

1. Line 52: a minor typo "if" → "of"

Authors' response: We have now corrected this typo

2. It's very hard to see the microbial composition from the supplementary figure S2. It would be very helpful if it could be provided as a separate table.

Authors' response: We have now increased the resolution of supplementary figure S2

Responses to reviewers' 2nd comments

Reviewer #1 (Remarks to the Author):

Reviewer #1 Final Statement

The authors have carefully and comprehensively addressed all of my previous comments. The additional methodological details, clarifications, and supplementary analyses provided have significantly improved the scientific soundness, reproducibility, and overall clarity of the manuscript. I also appreciate the inclusion of the suggested reference and the expanded discussion of potential confounders such as diet, hygiene, and seasonality. The revision substantially strengthens the study and its conclusions. I commend the authors for their thorough and thoughtful responses and have no further comments.

Reviewer #2 (Remarks to the Author):

I thank the authors for attempting to address the review comments. Concerns I raised regarding descriptions of biospecimen handling, and the clarity of Fig. S2 have been addressed well. However, I have two concerns for the revised version of this manuscript:

Remark 1: The added discussion about microbe-cytokine associations is severely lacking. The link between campylobacter and IL10 is well appreciated (IL10 knockouts are a standard mouse model for symptomatic *C. jejuni* infection), which has not been well communicated in the text. The following paragraph discusses literature on *Intestinibacter*, which is distinct from *Intestinimonas* and *Clostridium*, and so it is not clear why this is relevant. In my view, a major strength of a study with this design (microbiome and pathogen load/immune parameters assessed in the same individuals) is the ability to find these associations - contextualizing these should be a considerable focus of the discussion.

Authors' response: We have now added discussion about the microbe-cytokine associations discovered including the link between IL-10 and *C. jejuni* (page 9) and edited this section according to reviewer's comment.

Remark 2: I also find that the study has little relation to immune-mediated diseases as these are not included in any of the analyses, nor any specific biomarkers of these (e.g. autoantibodies), and any links to immune related diseases appear speculative. The Finnish cohort also appears to exclude children with HLAs associated with T1D, which will create bias in the data. Justification that HLA types do not impact cytokine levels is incorrect, as multiple studies have shown these to be linked, and also with inflammatory

diseases (PMID: 39697331, PMID: 40572779, PMID: 32127039). The findings the study has made are not appropriately contextualized with knowledge on the etiology of these diseases. Thus, I do not believe this study contributes any specific finding related to immune-related diseases - it is in my view inappropriate to reference immune-related diseases in the title of this manuscript.

Authors' response: We have now changed the title of the manuscript to "Early-life immunological and microbial differences between East African and North European children". We also modified the text discussing HLA types and cytokine levels (page 11).

Remark 3: The introduction states: "Despite this possible connection little research is done on early-life immune status, microbiota and microbial infections" - I do not believe this is the case as there is a strong body of literature linking these ideas e.g. PMID: 40482668 (, PMID: 35550670, PMID: 40175554, PMID: 37138015, PMID: 31964813, PMID: 31719945, PMID: 29602225, PMID: 24637604, PMID: 36419421, PMID: 37764797 etc

Authors' response: We have now clarified the sentence in the introduction as follows "Despite this possible connection little research is done on where early-life immune status, microbiota and microbial infections are analyzed simultaneously and consecutively in longitudinally followed children who live in contrasting microbial environments."

Reviewer #3 (Remarks to the Author):

The authors presented a longitudinal study comparing the trajectories of microbiome development of an African and a Western population. The reviewer's concerns were regarding the computational and statistical methods and were clarified by the authors in the changes that were made in the main text and the Methods section. The reviewer thus does not have follow up comments.

Responses to reviewers' 3rd comments

Reviewer #2 (Remarks to the Author):

Remark 1: The authors have made substantial efforts to address concerns raised in previous reviews - including acknowledging limitations of the study, removing the titular focus on immune-mediated diseases and improving discussion of findings within the current literature. In general, my view is that discussion of immune-mediated diseases is still a little heavy given lack of data on this aspect. Additionally, a major advantage of this study appears to be the longitudinal sampling, however findings related to this aren't really novel (e.g. its already known that the microbiota alpha diversity changes across these timepoints). Earlier I suggested including changes over time in the analyses between the populations - i.e. are there differences between the populations in how the microbiota or innate immune landscape change over time? - which I believe would strengthen the paper. The introduction/abstract mentions that 'characteristics of the immune system' were measured, which sounds more comprehensive than what was actually measured (four innate immune cytokines). e.g. "the characteristics of the immune system" - perhaps just change to "cytokines" or "innate immune cytokines"?

Authors' response: We have now edited the text to give less emphasis on immune-mediated diseases (Abstract, page 12 and page 18).

We added linear mixed effects regression models to analyze longitudinal change in the cytokine concentrations and gut alpha diversity (Page 9-10, Supplementary Table 2 and 3 and Supplementary Figures S2 and S4).

The phrase 'characteristics of the immune system' (introduction/abstract) has now been modified as suggested by the reviewer

Remark 2: Some editing is needed across the manuscript to avoid awkward-sounding phrasing - e.g. "Many diseases linked to immune system" should read "Many diseases linked to the immune system"; 'lifestyles' is more appropriate than 'lifestyle' in many cases.

Authors' response: These phrasings have now been modified

Remark 3: Two consecutive sentences appear in the introduction that begin with "Using longitudinal analysis and machine learning, the current study aims to explore overall and temporal differences in..."

Authors' response: The extra copy of the sentence has now been removed from the manuscript.